# Early Empirical Antibiotic Therapy Modification in Sepsis Using Beta-Lacta Test Directly on Blood Cultures

**Assaf Mizrahi [1,2], Françoise Jaureguy [3], Héloise Petit [3], Gauthier Péan de Ponfilly [1,2], Etienne Carbonnelle [3], Alban Le Monnier [1,2], Jean-Ralph Zahar [3,4,*] and Benoît Pilmis [2,5]**

1   Service de Microbiologie Clinique, Groupe Hospitalier Paris Saint-Joseph, 75014 Paris, France
2   Institut Micalis UMR 1319, INRAe, Université Paris-Saclay, AgroParisTech, 92290 Châtenay Malabry, France
3   Service de Microbiologie Clinique, Hôpitaux Universitaires de Paris Seine Denis, site Avicenne, AP-HP, 93000 Bobigny, France
4   UMR 1137, IAME, Université Paris, Sorbonne Paris Cité, 75018 Paris, France
5   Équipe Mobile de Microbiologie Clinique, Groupe Hospitalier Paris Saint-Joseph, 75014 Paris, France
*   Correspondence: jeanralph.zahar@aphp.fr; Tel.: +33-1-48-95-56-12

**Abstract:** *Background*: Sepsis caused by multi-drug-resistant Gram-negative bacilli lead physicians to prescribe broad-spectrum antibiotic therapy, such as carbapenems. Rapid susceptibility testing can help with the rational use of antibiotics. The aim of this study was to measure the clinical impact associated with rapid reporting of Beta-Lacta test (BLT) directly on blood cultures positive with Gram-negative bacilli. *Methods*: In an observational, multicentric, prospective study, we included patients with sepsis caused by Enterobacterales observed on Gram staining of the positive blood cultures. BLT and antimicrobial susceptibility testing (AST) were performed directly on the blood cultures. Clinical impact was measured on the proportion of patients for whom the probabilistic antibiotic therapy was modified according to BLT, including patients receiving carbapenem. *Results*: 170 patients were included, of whom 44 (25.9%) were receiving inadequate empirical antibiotic therapy. Among them, 27 (15.9%) benefited from an early modification, according to the BLT results. Among 126 (74.1%) patients receiving appropriate probabilistic antibiotic therapy, we modified the antibiotic therapy for 28 (16.5%) of them, including 4/14 (28.5%) de-escalation from carbapenem to a third-generation cephalosporin. *Conclusions*: Implementation of BLT performed directly on blood cultures allowed us to rapidly modify the empirical antibiotic therapy for about one-third of patients with sepsis caused by Enterobacterales.

**Keywords:** rapid diagnostic testing; ESBL; bloodstream infections

## 1. Introduction

The Infectious Diseases Society of America/Society for Healthcare Epidemiology of America (IDSA/SHEA) guidelines for implementing an antimicrobial stewardship program endorse the use of rapid diagnostic testing to assist in earlier identification of bacterial species and susceptibility in blood cultures. This recommendation is supported by the literature demonstrating that earlier time to de-escalation and initiation of effective antimicrobial therapy, decreased hospital length of stay, and decreased mortality when rapid diagnostic testing is utilized [1]. Numerous studies have demonstrated a correlation between mortality and time to therapy in Gram-negative bloodstream infections (BSI) [2–4]. Due to increased rates of multi-drug-resistant Gram-negative-bacilli-related infections, broad-spectrum antibiotic therapy is routinely prescribed especially for patients with healthcare-associated infections. Adaptation of the empirical therapy is performed after the results of antibiotic susceptibility testing (AST), which can take up to 48 h [5]. In clinical microbiology laboratories, the disc diffusion method described by Bauer et al. in 1966 remains one of the most frequently used AST methods [6]. Conventional AST methods performed directly on positive blood cultures have been recently optimized to obtain

reliable results in 6 h [7]. New AST methods, such as multiplex PCR assays, microarrays, morphokinetic cellular analysis, are emerging, but strong evidence about their clinical impact on patient management is still missing [8].

Rapid colorimetric assays can also help to determine some resistance mechanisms. The Beta-Lacta test (BLT) is a chromogenic test initially developed to detect third generation cephalosporin-resistant isolates from agar culture media [9]. Some studies showed that BLT could be used directly on blood cultures with a sensitivity ranging from 84.8% to 100% for the detection of Extended Spectrum Beta Lactamase (ESBL)-producing Enterobacterales [10–12]. We showed that BLT performed directly on blood cultures detected 100% of the ESBL-producing strains and could have allowed an antibiotic adaptation 28 h before the AST results [13].

Since this pilot study, BLT is performed in daily practice on positive blood cultures with Gram-negative bacilli on direct examination in our hospitals.

Therefore, the aim of this study was to measure the clinical impact associated with rapid reporting of BLT directly on blood cultures positive with Gram-negative bacilli combined with the antimicrobial stewardship team intervention.

## 2. Patients and Methods

### 2.1. Study Design and Patients

We conducted an observational, multicentric prospective study at the Saint-Joseph hospital, Paris and Avicenne hospital, Bobigny. Eligibility criteria were the presence of an active infection defined as a positive blood culture and clinical definition of sepsis caused by Enterobacterales according to Gram staining and requirement for an empirical antibiotic therapy [14]. Exclusion criteria were (i) infections related to bacteria other than Enterobacterales (ii) moribund patient or patient in palliative care for whom the clinician decided not to introduce antibiotic therapy. The study was carried out in accordance with the Declaration of Helsinki. This study was a non-interventional study with no addition to standard care. Biological material and clinical data were obtained only for standard diagnostic following physicians' prescriptions (no specific sampling, no modification of the sampling protocol). Data analyses were carried out using an anonymized database.

### 2.2. Microbiological Procedures

2.2.1. Blood Cultures

For the Saint-Joseph hospital, blood cultures were collected in BacT/ALERT bottles and incubated in Virtuo (bioMérieux, Marcy-l'Étoile, France). For Avicenne hospital, BD BACTEC™ Plus was used for collection of blood cultures that were incubated in BD BACTEC™ FX. Once the blood culture was flagged as positive, Gram staining was performed, as well as a BLT and a direct AST.

2.2.2. BetaLacta Test (BLT)

BLT was performed as previously described [13]. Briefly, positive blood culture broths were centrifuged at $5000 \times g$ for ten minutes. The supernatant was discarded, and bacteria were picked up at the surface from the gel tube with a 1 μL loop, which was finally emptied in the micro-tube containing one drop of both extraction and chromogenic reagents. All changes in the initial yellow color were interpreted as positive. We compared BLT results to antimicrobial susceptibility test results, and we determined sensitivity, specificity, as well as positive and negative predictive values of the BLT.

If there was a discrepancy between BLT and antimicrobial susceptibility test results, BLT was performed directly on colonies in solid media.

### 2.3. Antimicrobial Susceptibility Testing

Antimicrobial susceptibility testing was performed as previously described [13] by the disk diffusion method on Mueller Hinton agar (Biorad, Marnes-la-Coquette, France), according to the EUCAST guidelines [15]. All isolates showing reduced susceptibility to

CAZ 10 μg (zone diameter $\leq$ 22 mm and/or MIC $\geq$ 4 mg/L) and/or CTX 5 μg (zone diameter $\leq$ 20 mm and/or MIC $\geq$ 2 mg/L) were selected for ESBL/AmpC bêta-lactamase (chromosomally overproduced or plasmid-mediated) detection.

ESBL-producing bacteria were detected by the double-disk synergy test when bacteria were susceptible to cefoxitin (10 μg) [16]. Among AmpC-naturally producing Enterobacterales (*i.e., Enterobacter cloacae, Enterobacter aerogenes, Citrobacter freundii, Morganella morganii*) the combination disk method was also applied (cefepime + clavulanate disk (30 μg/10 μg) versus a cefepime disk (30 μg) alone) [17].

Phenotypic AmpC confirmation testing was considered positive when a restauration of cefotaxime or ceftazidime diameter was observed on Cloxacillin-supplemented Mueller–Hinton (Biorad) [18].

All Enterobacterales suspected to produce a carbapenemase were investigated according to the EUCAST guidelines [15].

Antimicrobial Adaptation Strategies according to the BLT

A standardized form according to antibiotic prescription was used to collect the data. Initial antibiotic treatment (defined as empiric treatment) was reported.

The results of BLT were reported to the antimicrobial stewardship team, which is composed of infectious disease specialists and clinical microbiologists. Each day, from 8:30 AM to 5:30 PM on weekdays and 8:30 AM to 1:00 PM on weekends, the antimicrobial stewardship team received in real time the information from the microbiology laboratory. Gram stain of positive blood cultures together with BLT results are discussed, and if necessary, the antimicrobial stewardship team modified the therapeutic decision.

Appropriate antibiotic treatment was defined as an efficient treatment tested in vitro against the isolated bacteria. When the patient received a combination of antibiotics (e.g., ceftriaxone plus amikacin), we considered that the antibiotic treatment was appropriate if one of both molecules was active on the isolated bacteria. By considering the BLT results only, we adapted the antibiotic therapy as follows: A carbapenem was chosen if the BLT was positive, and cefotaxime or ceftriaxone if the BLT was negative.

Definitions of escalation and de-escalation were based on the consensus published by Weiss et al. [19].

## 3. Results

### 3.1. Characteristics of Study Population

During the study period, 170 patients were included. Among these patients (M/F ratio = 1/1), the median age [IQR] was 71 (58–80) years. Source of infection were mainly urinary tract infections (*n* = 81; 47.7%) and intra-abdominal infections (*n* = 41; 24.1%) (Table 1). Among the 170 infections, 83 (48.8%) were considered healthcare-associated infections. Patient characteristics are presented in Table 1.

**Table 1.** Baseline characteristics of patients and bacterial identification.

|  | All Patients (*n* = 170) |
| --- | --- |
| *Demographic data* |  |
| Male/female, *n (%)/n (%)* | 81 (48)/89 (52) |
| Age (years), median (IQR) | 71 (58–80) |
| ICU admission, *n (%)* | 13 (7.6) |
| Healthcare-associated infections, *n (%)* | 83 (48.8%) |
| *Sources of infection, n (%)* |  |
| Urinary tract | 81 (47.7%) |
| Intra-abdominal | 41 (24.1%) |
| Primary bacteremia | 12 (7.1%) |
| Catheter-related | 11 (6.5%) |

**Table 1.** *Cont.*

|  | All Patients (*n* = 170) |
|---|---|
| Respiratory tract | 11 (6.5%) |
| Skin and soft tissue | 6 (3.5%) |
| Febrile neutropenia | 5 (2.9%) |
| Maternal-fetal | 1 (0.6%) |
| Neuromeningeal | 1 (0.6%) |
| Bone and joint infection | 1 (0.6%) |
| Monobacterial infection, *n (%)* | 158 (93) |
| Polybacterial infection, *n (%)* | 12 (7) |

### 3.2. Bacteriological Data

Distribution of Enterobacterales was as follows: 104 (61.2%) Escherichia coli, 40 (23.5%) Klebsiella sp., 13 (7.6%) Enterobacter cloacae complex, 4 (2.4%) Salmonella sp., 3 (1.7%) Proteus mirabilis, 2 (1.2%) Citrobacter freundii, 2 (1.2%) Citrobacter koseri, 1 (0.6%) Serratia mascescens, 1 (0.6%) Morganella morganii. One hundred fifty-eight patients (93%) had monomicrobial infection (Table 2).

**Table 2.** Resistance mechanisms of Enterobacterales and performance of the BLT.

| Enterobacterales Species | Total, *n* (%) | 3GC-S Strains | 3GC-R Strains | | |
|---|---|---|---|---|---|
| | | | ESBL | *AmpC* | Positive BLT, *n* (%) |
| **Groups 0, 1 and 2** | | | | | |
| *Escherichia coli* | **104 (61.2)** | 83 | 17 * | 4 | 16 (76.2) |
| *Klebsiella pneumoniae* | **35 (20.5)** | 30 | 5 ** | 0 | 3 (60) |
| *Klebsiella oxytoca* | **3 (1.8)** | 3 | 0 | 0 | 0 (0) |
| *Klebsiella variicola* | **2 (1.2)** | 1 | 1 | 0 | 1 (100) |
| *Proteus mirabilis* | **3 (1.7)** | 3 | 0 | 0 | 0 (0) |
| *Citrobacter koseri* | **2 (1.2)** | 2 | 0 | 0 | 0 (0) |
| *Salmonella spp* | **4 (2.4)** | 4 | 0 | 0 | 0 (0) |
| **Group 3** | | | | | |
| *Enterobacter cloacae* | **13 (7.6)** | 8 | 1 | 4 | 0 (0) |
| *Morganella morganii* | **1 (0.6)** | 1 | 0 | 0 | 0 (0) |
| *Citrobacter freundii* | **2 (1.2)** | 2 | 0 | 0 | 0 (0) |
| *Serratia marcescens* | **1 (0.6)** | 1 | 0 | 0 | 0 (0) |
| **Total** | **170** | **138** | **24** | **8** | **20 (62.5)** |

\* *E. coli* was ESBL + AmpC and counted in ESBL group. \*\* *K. pneumoniae* was ESBL + NDM carbapenemase.

### 3.3. Microbiological Performance of the BLT

ESBL-producing Enterobacterales were isolated in 24 (14%) cases, AmpC-hyperproducing Enterobacterales in eight (4.7%) cases, association of AmpC and ESBL was found in one (0.6%) case and one (0.6%) case of ESBL production associated with New-Delhi Metalloprotease (NDM) carbapenemase (Table 2).

In total, 32 strains were resistant to 3GC, and BLT detected 20 of them (62.5%). Performance of BLT for 3GC-resistant strains was calculated as follow: Sensitivity = 62.5%, Specificity = 100%, PPV = 100% and NPV = 92% (Supplementary Table S1).

When focusing on the 24 strains resistant to 3GC by ESBL production, 19 were detected directly on the blood cultures. Performance of BLT for ESBL-producing Enterobacterales was calculated as follow: Sensitivity = 79.2%; Specificity = 99.32%; PPV = 95%; NPV = 96.7% (Supplementary Table S1).

Regarding the five ESBL-producing strains not detected by the BLT directly on blood cultures, BLT was performed on the colony and was positive.

### 3.4. Empirical Antibiotic Therapy

Patients were empirically treated with third-generation cephalosporins (ceftriaxone, cefotaxime) (*n* = 86, 50.6%), β-lactam/β-lactamase inhibitor (*n* = 30, 17.6%), carbapenems

(*n* = 14, 8.2%), penicillins (*n* = 13; 7.6%), cefepime (*n* = 8, 4.8%), fluoroquinolons (*n* = 4, 2.4%) and β-lactam/β-lactamase inhibitor + fluoroquinolons (*n* = 2; 1.2%). Furthermore, thirteen patients (7.6%) were not treated with empirically antibiotic therapy. Twenty-five patients (14.7%) received aminoglycosides as empirical dual antibiotic therapy. Seventy-four percent (126/170) of patients received adequate empirical antibiotic therapy.

*3.5. Clinical Impact of BLT*

Of the 170 patients included, 44 (25.9%) were receiving inadequate probabilistic antibiotic therapy. Among them, 27 (15.9%) benefited from an early modification according to the BLT results, and 14 (8.2%) of the remaining 17 patients benefited from a modification of treatment according to the results of the antibiotic susceptibility testing (Figure 1).

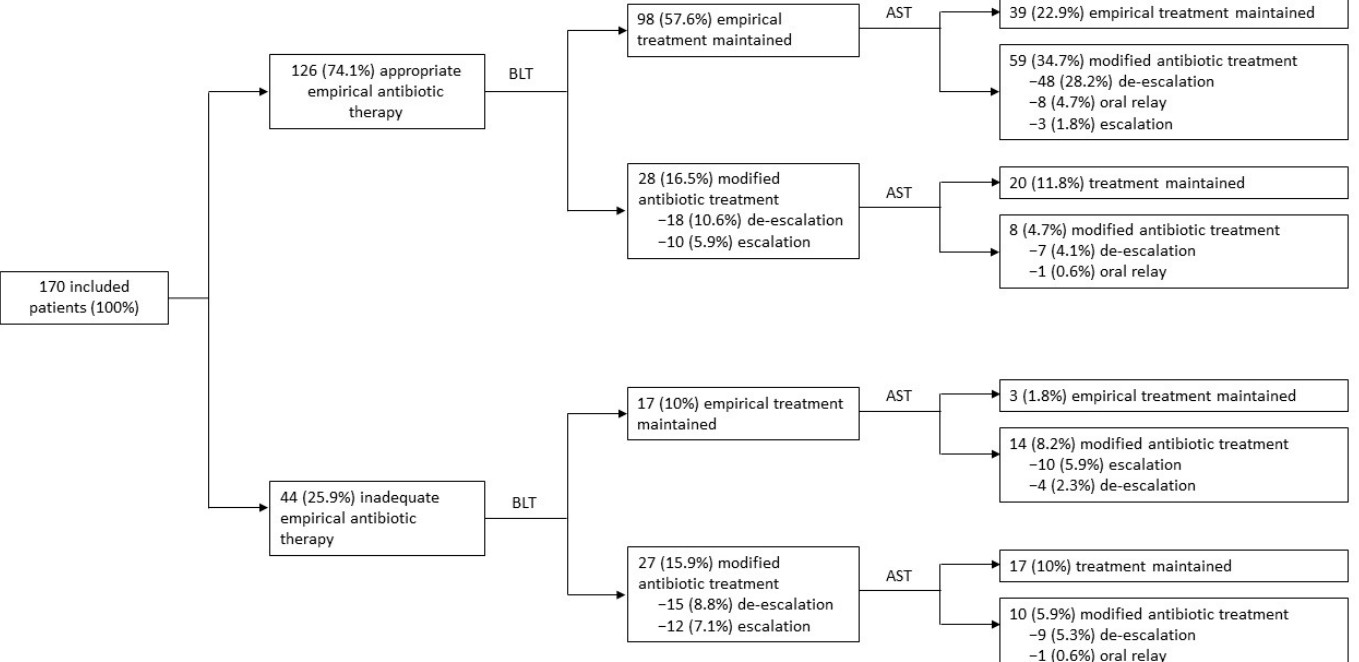

**Figure 1.** Adaptation of empirical antibiotic therapy according to BLT and AST.

Among 126 (74.1%) patients receiving appropriate probabilistic antibiotic therapy, we modified the antibiotic therapy for 28 (16.5%) of them, including 18 (10.6%) de-escalation and 10 (5.9%) escalation.

Among the 14 patients that received a carbapenem as an empiric antibiotic treatment, BLT was negative for half of them, which allowed us to de-escalate for a 3GC for four patients (28.5%).

Of the 20 patients with a positive BLT, 12 of them had inadequate empiric antibiotic therapy (60%). The BLT results allowed us to modify this EAT for 10 of them (50%), including seven escalations to a carbapenem (35%).

## 4. Discussion

The increasing incidence of infections related to multidrug-resistant Enterobacterales (by hyperproduction of AmpC, ESBL, or carbapenemase) is a real public health problem. Some authors reported that inappropriate initial antimicrobial therapy for septic shock occurs in about 20% of patients and is associated with a fivefold reduction in survival [20]. This observation often leads the clinician to prescribe broad-spectrum antibiotics (carbapenems) in patients at high risk of MDR infection or in known colonized patients. Furthermore, several studies showed that reduced time for bacterial identification and antimicrobial susceptibility testing using rapid tests is correlated with improved mortality and reduced healthcare costs. The impact of this rapid resistance identification and adjustment in em-

piric therapy is a major challenge. Thus, as reported by Blascke et al., rapid availability of accurate results from AST is currently considered one of the most important unmet medical needs in the management of infectious diseases. The BLT test, which is a rapid and inexpensive test, has been little evaluated in real life, especially in association with an antimicrobial stewardship team.

The clinical impact of BLT on bloodstream infection management has been evaluated in two studies. Garnier et al. evaluated an early antimicrobial therapy adaptation strategy guided by the BLT for ICU patients in a case-control study [21]. They included 61 patients in the BLT-guided adaptation strategy group, including 10% with a BSI. They showed by a multivariate analysis that the use of BLT was strongly associated with early appropriate and optimal antimicrobial therapies. Another study has been conducted by Dépret et al. on the use of BLT for the management of Gram-negative bacillary (GNB) bloodstream infections [22]. They evaluated if the BLT result would modify the antibiotic therapy proposed by an infectious diseases specialist. Contrary to Garnier et al., they conclude that BLT did not lead to a significant reduction in carbapenem prescription.

In our pilot study, we evaluated the theoretical impact of BLT on antibiotic therapy adaptation. BLT was performed on 141 blood cultures with Gram-negative bacilli, revealing 28 3GC-resistant bacteria (19.9%). Twenty-one patients (75%) received a non-adapted first-line treatment. The antibiotic therapy adaptation was delayed by 28.1 h, compared to the theoretical adaptation with BLT result.

In our present work, we have demonstrated that the rapid method of BLT-Test directly from blood culture allows effective modification of empirical antibiotic therapy.

Our study, carried out in a country with a low ESBL endemic, has shown that the use of the BLT leads to a modification of empirical antibiotic therapy in one-third of cases, particularly in terms of early de-escalation of broad-spectrum antibiotic therapy.

BLT failed to detect five ESBL-producing Enterobacterales directly on the blood cultures. Our hypothesis that could explain this lack of sensitivity is that the laboratory technicians did not collect enough or no bacteria for the test. Following these results, we therefore, set up routinely an inoculum control for all negative tests. The control consisted of the observation under the microscope of the Enterobacterales in the pellet used for the BLT. Nevertheless, BLT was positive for these five false negatives samples when performed directly on the colony, which would have led to a sensitivity of 100% for ESBL detection. It is to note that BLT sensitivity can differ according to the ESBL type. Indeed, Poirel et al. showed that the sensitivity of BLT for detecting non-CTX-M ESBL producers such as TEM or SHV was lower (84%) than for CTX-M producers (91%) [23]. A microbiological study evaluating the performances of BLT toward all ESBL and AmpC enzyme-producing strains with DNA sequencing analysis should be carried out in order to clarify precisely the microbiological performances of BLT.

Moreover, BLT did not seem to be suitable for the detection of resistance to third generation cephalosporins by AmpC hyperproduction. Following our previous study, which focused on the optimization of the BLT for the detection of ESBL-producing bacteria directly in urine samples, we decided to set up in daily basis a 30-min read of the BLT (instead of 15 min) as well as consider any change in the initial color as positive [24]. These modifications allowed to regain sensitivity for the detection of AmpC hyperproduction strains without losing specificity (data not shown). Furthermore, infections related to chromosomally carrier Enterobacterales are mainly healthcare-associated infections. Infectious diseases specialists and clinical microbiologists should keep in mind this lack of sensitivity and therefore advise cefepime for patients with healthcare-associated infections, moreover, if they are severe or if they received antibiotics recently before the BSI episode. The advantage of BLT comes from the fact that in our structures, the information is 100% utilized by trained infectious diseases and microbiology specialists. As reported by Messacar et al. "For a rapid diagnostic test to have a rapid impact, it is essential that providers be alerted to results in real time".

Our study has several limits, first, the clinical impact could depend on the working hours of the different labs.

The impact of this rapid diagnostic test on therapeutic decisions and adjustments in empirical antibiotic therapy remains a major challenge, and large prospective multicenter studies are needed to complete and confirm our results.

### 5. Conclusions

In this pilot study, implementation of BLT performed directly on blood cultures allowed us to modify the empirical antibiotic therapy for about one-third of patients (55/170) with sepsis caused by Enterobacterales. BLT allowed us to readjust rapidly the empirical carbapenem therapy, both for escalation or de-escalation. To avoid false negatives, it is essential to implement a control on the negative BLT. Microbiologists and infectious diseases specialists should also keep in mind the lack of sensitivity of BLT toward AmpC hyperproduction and advise cefepime for severe patients and/or nosocomial patients with previous recent antibiotic therapy. These results should be confirmed in a multicentric prospective study with a 24/24 implementation of BLT with and without an antimicrobial stewardship team.

**Supplementary Materials:** The following supporting information can be downloaded at: https://www.mdpi.com/article/10.3390/ijtm2030034/s1, Table S1: Baseline characteristics of patients and bacterial identification.

**Author Contributions:** Conceptualization, methodology, validation, investigation, writing—review and editing A.M., F.J., E.C., A.L.M., J.-R.Z., B.P. and G.P.d.P.; data curation, resources, H.P. and G.P.d.P.; writing—original draft preparation, A.M. and B.P. All authors have read and agreed to the published version of the manuscript.

**Funding:** This research received no external funding.

**Institutional Review Board Statement:** The study was carried out in accordance with the Declaration of Helsinki. This study was a non-interventional study with no addition to standard care. Biological material and clinical data were obtained only for standard diagnostic following physicians' prescriptions (no specific sampling, no modification of the sampling protocol). Data analyses were carried out using an anonymized database.

**Informed Consent Statement:** A no objection agreement was obtained from all subjects involved in the study.

**Data Availability Statement:** Not applicable.

**Conflicts of Interest:** The authors declare no conflict of interest.

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
