# Peer review of "Early Empirical Antibiotic Therapy Modification in Sepsis Using Beta-Lacta Test Directly on Blood Cultures"

_2673-8937, doi:10.3390/ijtm2030034_

Round 1
Reviewer 1 Report
Mizrahi et al. present the results of a prospective, observational study that utilized the Beta-Lacta test to inform antibacterial modification for septic patients with bacteremia caused by Enterobacterales. Antimicrobial stewardship is an important component of the international community’s effort to prevent the proliferation of antibacterial resistance, and the contents of the current investigation will likely be of interest to institutions that are looking for an affordable way to quickly identify the susceptibility of pathogens in the blood. I have provided constructive feedback for the authors that I believe will elevate the quality of the manuscript.
1) In the abstract, the acronym BLT, AST, and 3GC are not clearly defined.
2) The first comma in line 39 should be removed (and possibly the second comma?), and the comma in line 89 should be removed as well.
3) Line 47 – automated microdilution is one of the most prevalent forms of antimicrobial susceptibility testing as well and merits being mentioned.
4) Line 54 – cephalosporin-resistant should be hyphenated
5) Line 69 – can the authors please expand on the “clinical definition of sepsis?” Is sepsis defined using SOFA scores per the Sepsis-3 definition, SIRS criteria, or a different definition?
6) Lines 99 - 100 – The authors use the terms CAZ and CTX without defining the acronyms and then later use the full words cefotaxime and ceftazidime in line 109. I recommend spelling out the entire drug names to reduce the number of acronyms and to keep the terminology consistent.
7) Line 105 – Enterobacter aerogenes was reclassified as Klebsiella aerogenes (same comment for Table 2)
8) Line 108 – I believe the authors intended to use the word “restoration”
9) Lines 111 – 112, can the authors please elaborate on how the carbapenemase-producers were “investigated?” Something simple like listing the assay that was used to detect the presence of carbapenemase enzymes will provide needed clarity. In Table 2 one of the isolates was revealed to be an NDM-producer, so it sounds like the hospital is able to identify which enzymes are produced by carbapenamase producing organisms.
10) Line 118 – are the infectious diseases specialist on the stewardship team all physicians or is it a multidisciplinary team of clinicians?
11) Line 122 – when the authors state that the antimicrobial stewardship team modified therapeutic decisions, can the authors please elaborate on the logistics of how the patient’s antibacterial regimen were modified? Did the stewardship team contact the physician caring for the patient and recommend a therapeutic change, and if so, did the primary physician ever decline the stewardship team’s recommendation?
12) Line 136 – the Infectious Diseases Society of America has been slowly moving away from using the term healthcare-associated infections because of the overuse of broad spectrum antibacterials. As an example, the IDSA no longer recommends using the term healthcare-associated pneumonia. Can the authors please clarify which criteria were used to determine if an infection was “healthcare-associated” as opposed to the term “hospital acquired” which has a stricter definition of an infection that typically manifest greater than 48 hours after being admitted to the hospital?
13) Line 169 – can the authors please provide an explanation for why 7.6% of patients did not receive empiric antibacterials? Did the medical team believe the patient did not have an infection despite otherwise meeting the criteria for sepsis?
14) My largest suggestion to the authors is to discuss more about the strengths and drawbacks of the Beta-Lacta test in comparison to other assays, which I believe will better contextualize the results of their study. For example, commercially available rapid diagnostics that utilize PCR arrays are capable of detecting ESBL enzyme production from organisms like Escherichia coli and Klebsiella pneumoniae. In comparison to rapid diagnostics/resistance mechanism detection, what are the benefits of the Beta-Lacta test? The authors admit in line 238 that the Beta-Lacta test struggled to detect overproduction of AmpC, so if hospitals currently have PCR based assays that detect ESBL production is there a benefit to using the Beta-Lacta test, and if so, in which situations?
15) If there is room in the abstract I think the authors should include the sensitivity, specificity, PPV, and NPV of the Beta-Lacta test for detecting Enterobacterales resistant to third generation cephalosporins to fairly summarize the utility of the assay. A sensitivity of 62.5% is something that clinicians intending to the use assay should be aware of so that they can take the necessary precautions outlined by the authors.
Reviewer 2 Report
I consider that it is an interesting and well organised study. I have no comments and suggestions.
Author Response
No modification were required.